# Climate targets, Carbon Dioxide Removal and the potential role of Ocean Alkalinity Enhancement

Andreas Oschlies[1], Lennart T. Bach[2], Rosalind Rickaby[3], Terre Satterfield[4], Romany Webb[5], Jean-Pierre Gattuso[6,7]

[1]GEOMAR Helmholtz Centre for Ocean Research Kiel, 24148 Kiel, Germany
[2]Institute for Marine and Antarctic Studies, University of Tasmania, Hobart, Australia
[3]Department of Earth Sciences, University of Oxford, Oxford, UK
[4] University of British Columbia, Vancouver, Canada
[5] Sabin Center for Climate Change Law, Columbia Law School, New York, US
[6] Sorbonne Université, CNRS, Laboratoire d'Océanographie de Villefranche, 181 chemin du Lazaret, F-06230 Villefranche-sur-Mer, France
[7] Institute for Sustainable Development and International Relations, Sciences Po, 27 rue Saint Guillaume, F-75007 Paris, France

*Correspondence to*: Andreas Oschlies (aoschlies@geomar.de)

**Abstract.** The Paris Agreement to limit global warming to well below 2°C requires ambitious reduction of greenhouse gas emissions and the balancing of remaining emissions through carbon sinks, i.e. the deployment of Carbon Dioxide Removal (CDR). While ambitious climate mitigation scenarios until now consider primarily land-based CDR methods, there is growing concern about their potential to deliver sufficient CDR, and marine CDR options are receiving more and more interest. Based on idealized theoretical studies, Ocean Alkalinity Enhancement (OAE) appears as a promising marine CDR method. However, the knowledge base is insufficient for a robust assessment of its practical feasibility, of its side effects, social and governance aspects as well as monitoring and verification issues. A number of research efforts aim to improve this in a timely manner. We provide an overview on the current situation of developing OAE as marine CDR method, and describe the history that has led to the creation of the OAE research Best Practices Guide.

## 1 Climate Goals and the need for Carbon Dioxide Removal

A key finding of climate research in recent decades is that the increase in global mean surface air temperature since the beginning of industrialization is proportional to cumulative emissions of carbon dioxide ($CO_2$), the major anthropogenic greenhouse gas (Matthews et al. 2009). The Paris Agreement's goal of limiting global warming to well below 2°C, and ideally 1.5°C above pre-industrial levels (UNFCCC, 2015) can thus be converted to a remaining carbon budget that, for current global emissions, will be used up in a few years for the 1.5°C target and about 2 decades for the 2°C target (United Nations Environmental Programme, 2022). The Paris Agreement thus explicitly demands ambitious reductions in anthropogenic greenhouse gas emissions and the balancing of hard-to-abate emissions through carbon sinks in the second half of the 21st

century (UNFCCC, 2015). The balance to be achieved is also called net zero and is a qualitatively new element compared to previous climate protection agreements.


Arresting global warming will require net-zero $CO_2$ emissions. Non-$CO_2$ greenhouse gases (GHGs), in particular nitrous oxide and methane, also contribute to current warming. However, because their lifetime in the atmosphere is considerably shorter than that of $CO_2$, arresting global warming does not require net-zero emissions for non-$CO_2$ GHG  (Allen et al., 2022). Nevertheless,  increases in non-$CO_2$ GHG  emissions may lead to further temperature rise, whereas a decrease in non-$CO_2$

GHG emissions will relatively quickly reduce atmospheric concentrations of the respective non-$CO_2$ GHG and thus radiative forcing and global warming. In order to achieve the long-term temperature goal, parties to the Paris Agreement agreed to reach global peaking of GHG emissions as soon as possible, to undertake rapid reductions thereafter, and to achieve a balance, i.e. net zero, between anthropogenic emissions by sources and removals by sinks of greenhouse gases in the second half of this century. The Paris Agreement adopts the UNFCCC definition of "sink," which refers to "any process, activity or mechanism

which removes a greenhouse gas . . . from the atmosphere," and thus encompasses both ecosystem-based and more technological or engineered removal approaches. Presently, no viable method exists for large-scale removal of non-$CO_2$ GHGs. Therefore, Carbon Dioxide Removal will likely have to balance not only hard-to-abate residual emissions of $CO_2$, e.g. from cement production, waste incineration, aviation and maritime transport, but also emissions of non-$CO_2$ GHGs, in particular from agriculture.


The amount of these residual emissions needs to be politically and socially viable. In principle, all 'hard-to-abate' emissions are technically avoidable, e.g. by switching from fossil to renewable energy, by capturing and safely storing $CO_2$ from process emissions (e.g. cement production using renewable energy), or by avoiding the processes that lead to emissions. Particularly in the agricultural sector, avoiding all emissions appears impossible without critical societal impacts: rice production and the

raising of livestock are associated with methane production, and any use of nitrogen fertilizer is associated  with nitrous oxide production, which are both potent non-$CO_2$ greenhouse gases. The exact amount of residual emissions is thus largely an issue of economic and social costs and society's  ambition to avoid emissions. Which emissions are deemed unavoidable also varies across historical and political contexts and is influenced by claims as to what is regarded as legitimately possible (Lund et al. 2023). Ultimately, decisions about the amount of residual emissions depend on values, norms and interests. Current scenarios

assume that, by mid-century, residual emissions will amount to between 10% and  20% of today's emissions, i.e. about 6 to 12 Gt $CO_2$e per year globally), where $CO_2$e includes the $CO_2$ equivalents of non-$CO_2$ GHGs that are estimated to contribute half to two thirds of the residual emissions (Buck et al., 2023).

Current global Carbon Dioxide Removal has been estimated near 2 Gt $CO_2$ per year almost exclusively by conventional management of land, primarily forest management (Grassi et al., 2021), afforestation and reforestation and with only 0.002 Gt

$CO_2$ $yr^{-1}$ by 'novel' CDR schemes comprising bioenergy with carbon capture and storage (BECCS), direct air capture with carbon storage (DACCS), enhanced weathering, and marine CDR, also sometime called ocean CDR, including ocean alkalinity enhancement (OAE) as a subcategory (Smith et al., 2023). According to the 'State of CDR' report (Smith et al., 2023), deployment of novel CDR approaches will have to increase by three orders of magnitude by mid century in order to reach net-zero emissions even in most ambitious emission reduction scenarios. Note that many scenarios used in the recent IPCC's 6th Assessment Report (IPCC, 2022) assume that emissions turn net negative after having reached net zero (Fig.1), which would allow a net cooling and is also deemed necessary for so-called temperature overshoot scenarios (Geden and Löschel, 2017) that receive more attention the longer it takes to drastically reduce emissions. It is, however, currently unclear how to incentivize and govern net negative emissions.

While the current climate goal of the UNFCCC is to limit the temperature rise to 2°C or 1.5°C relative to pre-industrial levels, one could also envisage climate targets that aim to reduce global temperatures further toward pre-industrial levels, and much faster than the tens to hundreds of millennia that planetary feedbacks would take to do so (Archer et al., 2009). Should humanity aim for a faster restoration of the planetary thermal balance to pre-industrial times, CDR would be a prime mechanism with deployment required well beyond the current 'net-zero' targets.

## 2 CDR approaches and the role the ocean could play

Traditionally, the focus of CDR has been on land-based methods such as reforestation and afforestation or BECCS. While these approaches certainly have some potential, there are unresolved issues related to land-use competition and associated political and societal feasibility challenges, and it is currently unclear if and how their combined deployment will be possible at scales sufficient to meet the net-zero target by mid century (The Land Gap Report, Dooley et al., 2022). It is thus unlikely that such terrestrial ecosystem-based solutions alone will be sufficient to achieve net-zero (Smith et al., 2023) and therefore 'novel' approaches will also have to be applied to a considerable extent. None of these are ready for large-scale deployment today. Transparent research into the efficacy, risks, and benefits of different approaches is urgently needed, and the societal debate on what counts as residual emissions and whether and how to deploy different CDR approaches must begin quickly so that appropriate processes can be developed in time, well-informed decisions can be made about research, development, and deployment, and mechanisms can be devised to regulate such use responsibly. Importantly, deployment at scale could compete with other societal demands for land, water and energy (Lawrence et al., 2018). Marine CDR has the potential to reduce the need for land and freshwater resources. Large-scale marine CDR approaches, however, may struggle to achieve public acceptance (Bertram and Merk 2020, Nawaz et al., 2023).

Marine CDR options are receiving more and more interest, acknowledging that the ocean has already absorbed more than a quarter of anthropogenic $CO_2$ emissions and would, on timescales of thousands to hundreds of thousands of years, take up

most of the remaining emissions (Archer and Brovkin, 2008), as it has done with natural high-CO₂ excursions in the Earth's geological past. The ocean holds more than 50 times as much carbon (primarily in the form of dissolved inorganic carbon) as the pre-industrial atmosphere and about 20 times as much as the carbon stored in global terrestrial plants and soils (Carlson et al., 2001). Its theoretical carbon storage potential appears large compared to the atmospheric and terrestrial carbon pools. However, increasing the oceanic carbon pool will affect the marine environment and may put additional pressure on marine ecosystems. The current level of scientific understanding of marine CDR is low, and more research is required to comprehensively assess the diverse portfolio of proposed options (e.g. NASEM, 2021). A particular challenge for marine CDR concerns monitoring and verification of any CDR-induced carbon fluxes and carbon storage, essential for reliable and honest carbon crediting (Boyd et al., 2023). Detection and attribution of CDR signals is particularly difficult due to the large natural marine carbon pool that already contains a considerable anthropogenic signal. The high temporal and spatial variability of these signals as well as the temporal and spatial decoupling of air-sea $CO_2$ fluxes and carbon storage in the interior pose specific challenges to detection and attribution of CDR. The determination of a baseline,  the additional carbon sequestered beyond the baseline, and quantification of carbon storage durability will likely be associated with considerable uncertainties. A key aspect of Monitoring, Reporting and Verification (MRV) is the development of transparent schemes that allow a reliable determination of CDR, and of consequent impacts on the carbon cycle and hence climate, as well as the association of carbon credits with individual CDR activities.

Currently considered marine CDR approaches include: (1) biological methods such as photosynthetic carbon fixation by microalgae, macrophytes (e.g. seaweeds) or mangrove trees and subsequent storage of carbon in the deep ocean or in coastal sediments, and (2) abiotic methods that aim to alter the carbonate chemistry of seawater in a way that enhances air-to-sea flux of $CO_2$ and subsequently stores atmospheric carbon as dissolved inorganic carbon in seawater. Hybrid biological, physical and/or chemical marine CDR approaches are also considered (artificial upwelling/downwelling, marine BECCS, bio-enhanced alkalinity generation, hybrid ocean-geochemical approaches, etc.). Among marine CDR methods investigated, abiotic approaches have been assessed as those with the lowest knowledge base and highest efficacy (Gattuso et al. 2018; NASEM, 2021). Improving their knowledge base therefore appears critical, and we focus in this OAE Guide 23 on ocean alkalinity enhancement.

## 3 Ocean Alkalinity Enhancement

Ocean alkalinity enhancement is a marine CDR concept with high theoretical sequestration potential in the range of 3 to 30 Gt $CO_2$ yr$^{-1}$ (Köhler et al., 2013; Renforth and Henderson, 2017; Feng et al., 2017), for which a number of technical deployment approaches have been suggested (Figure 2). Alkalinity, the excess of proton acceptors over donors, is a chemical concept that largely determines the storage capacity for $CO_2$ in seawater. OAE aims to enhance alkalinity by adding alkaline material to the surface ocean or by removing acid from seawater via electrochemistry. Alkalinity enhancement results in the consumption

of protons, a corresponding increase in the pH, which results in a decrease of the partial pressure of $CO_2$ in seawater. If applied to the surface ocean, and depending on the initial air-sea $CO_2$ gradient, it would promote $CO_2$ uptake from - or lessen $CO_2$ release to - the atmosphere, in both cases leading to a net reduction in atmospheric $CO_2$ at the expense of an increase in the oceanic pool of dissolved inorganic carbon. The atmospheric $CO_2$ absorbed via OAE-induced air-sea gas exchange is
essentially stored in the form of dissolved bicarbonate and carbonate ions that do not exchange with the atmosphere.

When applied to the surface ocean, OAE can rely on air-sea gas exchange to at least partially restore the OAE-induced decrease in the partial pressure of $CO_2$. Air-sea gas equilibration of $CO_2$ can take months to years (Jones et al., 2014) and may pose specific challenges to MRV (He and Tyka, 2023). However, along the path to equilibration, air-sea $CO_2$ fluxes approach zero
and would, for otherwise constant environmental conditions, follow an inverse exponential function, for which a disproportionate share of the total $CO_2$ flux occurs at the beginning of the equilibration period. The complex impacts of mixing and transport of water masses in reality make direct observations of the $CO_2$ influx unfeasible. Numerical models may be required for reliable quantification of air-sea gas exchange, whose skill has yet to be demonstrated (Bach et al., 2023). OAE can also be applied by adding alkalinity to chemical reactors upstream, that could at least partially pre-equilibrate the alkalized
seawater with additional $CO_2$ taken either from ambient air or from $CO_2$ waste streams. If $CO_2$ is taken from waste streams, this would, technically, correspond to emissions avoidance and not CDR. Also, if this $CO_2$ was taken from ambient air via, e.g. direct air capture facilities or bioenergy plants, CDR would be termed according to the process that removes additional $CO_2$ from the atmosphere and not the process that provides terminal carbon storage. OAE applied to chemical reactors or to the surface ocean qualifies as marine CDR if it leads to a net removal of $CO_2$ from the atmosphere, either by increasing the
flux of $CO_2$ from the atmosphere to the ocean or by reducing the emissions of $CO_2$ from the ocean to the atmosphere. Hybrid schemes that combine emission reduction by dissolving minerals with acidic $CO_2$ waste streams in chemical reactors to generate dissolved alkaline solutions to be added into the ocean for subsequent marine CDR can also be envisaged.

OAE was positioned in the "Concept Stage" cluster of a recent assessment of ocean-based measures for climate action (Gattuso
et al., 2021). This cluster was defined for measures with potentially very high effectiveness but with feasibility and cost-effectiveness which have yet to be demonstrated. The assessment highlighted the urgent need to improve knowledge on "Concept Stage" measures because the full implementation of proven measures runs the risk of falling short of providing enough cost effective CDR capacity. Attractive aspects of OAE compared to many other methods, in particular those that store carbon in biomass, are its potential to reduce ocean acidification at least locally (Albright et al., 2016), and the theoretical
durability of storage over several tens to hundreds of thousand years. An effective leakage of $CO_2$, either via enhanced $CO_2$ flux back to the atmosphere or by reduced $CO_2$ uptake from the atmosphere compared to a baseline scenario, can result from enhanced formation and reduced dissolution of carbonate minerals in the water column or at the sea floor. Possible leakage effects via impacts of OAE on pelagic calcifiers are uncertain (Bach et al., 2019), and feedbacks via changes in dissolution and preservation of carbonates on the sea floor operate on timescales of hundreds to thousands of years (e.g. Gehlen et al.,

2008). While there is little indication that leakage is a major concern for OAE on shorter than centennial timescales, a quantitative assessment of leakage across the spectrum of timescales is lacking. Frequently mentioned drawbacks of OAE are (i) the amount and the quality of alkaline material that is needed (whether mined in the case of mineral-based approaches or generated from waste brine in electrochemical approaches) and the energy required (whether mining, grinding, and transport for mineral-based approaches or the source of electricity for electrochemical approaches), and (ii) the difficulty of reliably

quantifying CDR (MRV). Regarding (i), all known CDR methods require, at climate-relevant scales, the movement of large amounts of matter. In addition to abundant carbonate and silicate minerals, a number of industrial waste products or artificial minerals can also be considered as alkalinity sources (Renforth, 2019, Caserini et al., 2022). Employing these for OAE would require proper accounting of any $CO_2$ emitted in their production (e.g., $Ca(OH)_2$ or $Mg(OH)_2$ produced through calcination of $CaCO_3$ or $MgCO_3$, respectively). Overall, there is no shortage of alkaline materials on the planet (Caserini et al., 2022).

Regarding (ii), MRV is indeed a challenge and is addressed by Ho et al. (2023, this Guide).

So far, the CDR potential of OAE has essentially been inferred from modeling and techno-economic studies (Kheshgi, 1995; Harvey, 2008), including spatially resolved global or regional models (e.g., Ilyina et al., 2013; Keller et al., 2014, Hauck et al., 2016; Wang et al., 2023). Such models employ simplified representations of marine biogeochemistry, rudimentary descriptions

of marine ecosystems, and typically simulate OAE as the addition, often instantaneously, of 'pure' alkalinity or of olivine minerals consisting of silicate, iron and alkalinity. Such studies can provide large-scale estimates of the theoretical CDR potential of OAE. Small-scale experimental studies can complement this with insight into realizable effectiveness of alkalinity addition and with the investigation of impacts that cannot be predicted from simplified modeled systems, such as environmental side effects. The first experimental studies have started only recently and have already generated novel insight into issues

regarding the actual delivery of alkalinity, in particular the risk of calcium carbonate precipitation that may negate intended CDR effects (Fuhr et al., 2022; Moras et al., 2022; Hartmann et al., 2023), and ecological impacts (Ferderer et al., 2022), and further research efforts are underway. Some information on the biogeochemical and ecological impacts of OAE might be gained from experimental work on ocean acidification that has been carried out during recent decades. Indeed, a first ocean OAE field experiment was carried out in the context of ocean acidification research. It used alkalinity addition to demonstrate

that ocean acidification is detrimental to coral reef calcification and that alkalinity addition can alleviate some effects of ocean acidification (Albright et al., 2016). Insight into possible impacts of OAE on marine organisms can be gained from research by the shellfish industry investigating the utility of so-called 'sweetening' the water through addition of mainly soda ash $(Na_2CO_3)$, a practice utilized in shellfish hatcheries for decades, and also in the academic and industrial fields of 'river liming', which dissolved primarily $CaCO_3$ and dolomite in higher latitude watersheds to offset the effects of acid rain in the 1960s and

1970, but is still practiced today in Canada and some Scandinavian countries, among other places (Mant et al., 2013).

Still, crucial knowledge gaps exist. Issues to be researched include the method of alkalinity addition, the alkaline materials to be used as well as their processing, the key attributes of ideal locations for deployment, the CDR potential that can be realized

on given timescales, durability of the carbon storage, biogeochemical and ecological co-benefits and risks, as well as MRV
and economic, legal, social, and ethical aspects of OAE. Of particular relevance for OAE and most other marine CDR methods is the regulatory perspective at international level. This is required to govern activities affecting the ocean as part of the global commons.

The very few (less then 10 according to the authors' knowledge) field trials that have been carried out so far, or are being discussed in the year 2023, have the potential to take up a few tons of $CO_2$ per trial. For the various OAE approaches, technology readiness levels (TRLs) are relatively low, generally rated as 1-2 by Smith et al. (2023), 3-4 for specific approaches (Foteinis et al., 2022) and possibly approaching 5 to 6 for methods with first field trials in preparation or under way (see Eisaman et al., 2023, this Guide). Scaling up $CO_2$ uptake by several orders of magnitude to many million tons per year or possibly even a billion tons per year by mid-century is extremely ambitious. It would require all instruments, measures and policies put in place that can advance every option forward from its current readiness level. In their State of CDR report Smith et al. (2023) estimate that so-called novel CDR methods, which include OAE, would need to be scaled up about by a factor 30 by 2030 and a factor of 1300 by mid-century in order to meet the demand expected for reaching promised climate goals. Required average annual OAE growth rates will have to be around 50%, which is extremely ambitious compared to, for example, an average 9% annual increase in the global capacity of renewable energy (IRENA 2021). Whether or not CDR and OAE specifically can be scaled up sufficiently by mid-century will depend on progress over the next decade, which Smith et al. (2023) call 'novel CDR's formative years'. A possible advantage of most OAE methods is that, technologically, they appear relatively simple and rely, to a substantial degree, on technology that exists already for processing different mineral resources at annual rates similar to those that may be required by OAE by mid-century. A possible roadblock for rapidly scaling up OAE may be a lack of public acceptance (Bertram and Merk, 2020; Nawaz et al., 2023).

In addition to technological challenges and acceptability issues that would need to be resolved, appropriate governance schemes will be needed if OAE is to be deployed at climatically relevant scales (GESAMP, 2019; Boettcher et al., 2021). The 2013 amendment to the London Protocol offers an approach for governing marine CDR, with a focus on ocean fertilization, but would need to be developed further with regards to OAE (see Steenkamp and Webb, 2023, this Guide). Interactions between OAE and other ocean-based activities will also need to be considered (e.g., via marine spatial planning), and any climate-relevant OAE deployment would require new or significantly expanded climate policies and financing schemes. Inclusion of OAE in carbon markets will require the establishment of robust MRV procedures.

All these issues need to be resolved before OAE can be implemented at large scale. Achieving this by mid-century is challenging, but not impossible. Research is urgently required on all aspects that are addressed in the various papers of the OAE Guide 23.

## 4 Motivation for developing a best practices guide

Given the urgency of establishing a portfolio of CDR options, a rapid improvement of our understanding of the carbon storage potential and of the co-benefits and risks of OAE is needed. This requires responsible, efficient and transparent scientific research in order to generate new and reliable information, allowing for rapid sharing, testing and synthesis of results. With the first publicly funded research projects having started in several countries, philanthropy funding a number of research projects to accelerate scientific progress, and start-ups working on enhancing technological readiness and developing scalable methodologies, this has motivated us to develop a best practices guide for OAE research.

The papers included in this guide describe current knowledge on the strengths and weaknesses of different OAE approaches, scientific uncertainties, biological and ecological impacts, knowledge gaps and research needs. Recommendations for experimental set-up of laboratory, pelagic and benthic mesocosms and field experiments, as well as for modeling approaches are provided. The guide also discusses the legal context in which research occurs and offers recommendations for responsible research and innovation, public engagement, data reporting and sharing, MRV and attribution.

The best practices guide aims at fostering intercomparison and synthesis efforts of different studies evaluating the potential, effectiveness and durability of OAE. This will help to improve knowledge sharing and information gain, and thereby speed up scientific progress at a time when robust information about OAE as a Carbon Dioxide Removal option is urgently needed to enable society to define and design appropriate actions to reach agreed climate goals.

This research field is in its infancy and is rapidly evolving. The broader legal and social contexts in which research occurs are also undergoing change. What we designate as "best practice" in this guide today may not be considered best practice in the future. As such, our guide comprises our current understanding onOAE, but it is critical that users remain up to date with the literature published after publication of the OAE guide 23. There will almost certainly be improvements in  protocols as the field develops and everyone is invited to contribute to this process.

## 5 Development of this best practices guide

Best practices guides have proven useful when new areas of research open up, often bringing together scientists from different fields and with different methodological backgrounds. One example is the Guide to Best Practices in Ocean Acidification Research and Data Reporting (Riebesell et al., 2010), in which the project lead, Jean-Pierre Gattuso, the scientific coordinator, Andreas Oschlies, and a number of authors of this guide were involved. The ocean acidification guide had an enormous catalytic effect in growing the field of ocean acidification research by lowering the barrier to entry and making comparison of different studies and the generation of synthesis products more straightforward. The expectation is that the present guide on

OAE research will have a similar impact on the OAE community and ocean CDR field at large, and also provide guidelines for ensuring that OAE research is conducted responsibly and most efficiently for the public good.

In summer 2022, Jean-Pierre Gattuso and Andreas Oschlies sent a proposal to the ClimateWorks Foundation with a request for funding to produce a detailed guide that outlines all the relevant approaches available for researching Ocean Alkalinity Enhancement as a Carbon Dioxide Removal method. The requested funding for a part-time project manager, a 3-day in-person workshop of chapter lead authors, as well as costs for production, publication and printing of the guide (total amount 170,000 US$) was approved. A steering committee consisting of the authors of the present paper was established and had several online meetings to develop outline and a conflict-of-interest form that all authors would have to sign in order to ensure transparency, and best scientific knowledge and the absence of conflicts of interest. Lead authors for each paper of the guide were chosen by the steering committee based on experience, scholarship, and diversity. In consultation with the steering committee, all lead authors then chose co-leads and additional authors of their respective papers.

In early 2023, a 3-day in-person workshop of the steering committee and lead authors took place in Villefranche-sur-Mer, France. All paper outlines were discussed, gaps identified and the timeline agreed upon. Lead authors were responsible for developing their papers, with support from the scientific project manager. A public website (https://oae-best-practice.carbondioxide-removal.eu ) with list of papers and lead-authors was set up and advertised via social media and the Carbon-Dioxide-Removal news stream (www.carbondioxideremoval.eu). An internal review was initiated in May 2023. All papers were submitted to '*State of the Planet*' in order to allow for public review to ensure that the OAE Guide 23 provides state-of-the-art information.

**Key recommendations**

- Research on ocean alkalinity enhancement should consider and, whenever appropriate, follow the best practices lined out in the OAE Guide 23
- Results of all experiments should be shared transparently, irrespective of whether experimental outcomes are considered 'positive' (e.g.. affirmative of the experimenters' prior assumptions), 'negative' or 'neutral'. This includes full transparency of OAE research that provides additional complications and/or roadblocks to OAE implementation.
- We recommend establishing a public registry for OAE field experiments, where all field experiments should be registered before the experiment is carried out.
- Researchers on OAE should help to further develop and improve the best practices lined out here and eventually strive for an update of the OAE Guide 23 in the future.

**Author Contributions**

AO conceived and all authors wrote and edited the paper

## Competing interests

The authors declare no competing interests

## Acknowledgements

This is a contribution to the "Guide for Best Practices on Ocean Alkalinity Enhancement Research". We thank our funders the ClimateWorks Foundation and the Prince Albert II of Monaco Foundation. Thanks are also due to the Villefranche Oceanographic Laboratory for supporting the lead authors' meeting in January 2023. We thank Miranda Boettcher, Kai Schulz, Matt Eisaman and Greg Rau for constructive comments on earlier versions of this manuscript. AO acknowledges funding from the European Union's Horizon 2020 Research and Innovation Program under grant 869357 (project OceanNETs: Ocean-based Negative Emission Technologies analyzing the feasibility, risks, and co-benefits of ocean-based negative emission technologies for stabilizing the climate) and from the German Federal Ministry of Education and Research (Grant No 03F0895) Project RETAKE, DAM Mission "Marine carbon sinks in decarbonization pathways" (CDRmare). We are grateful to Justin Ries and three anonymous referees for providing constructive comments that helped to improve the manuscript.

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

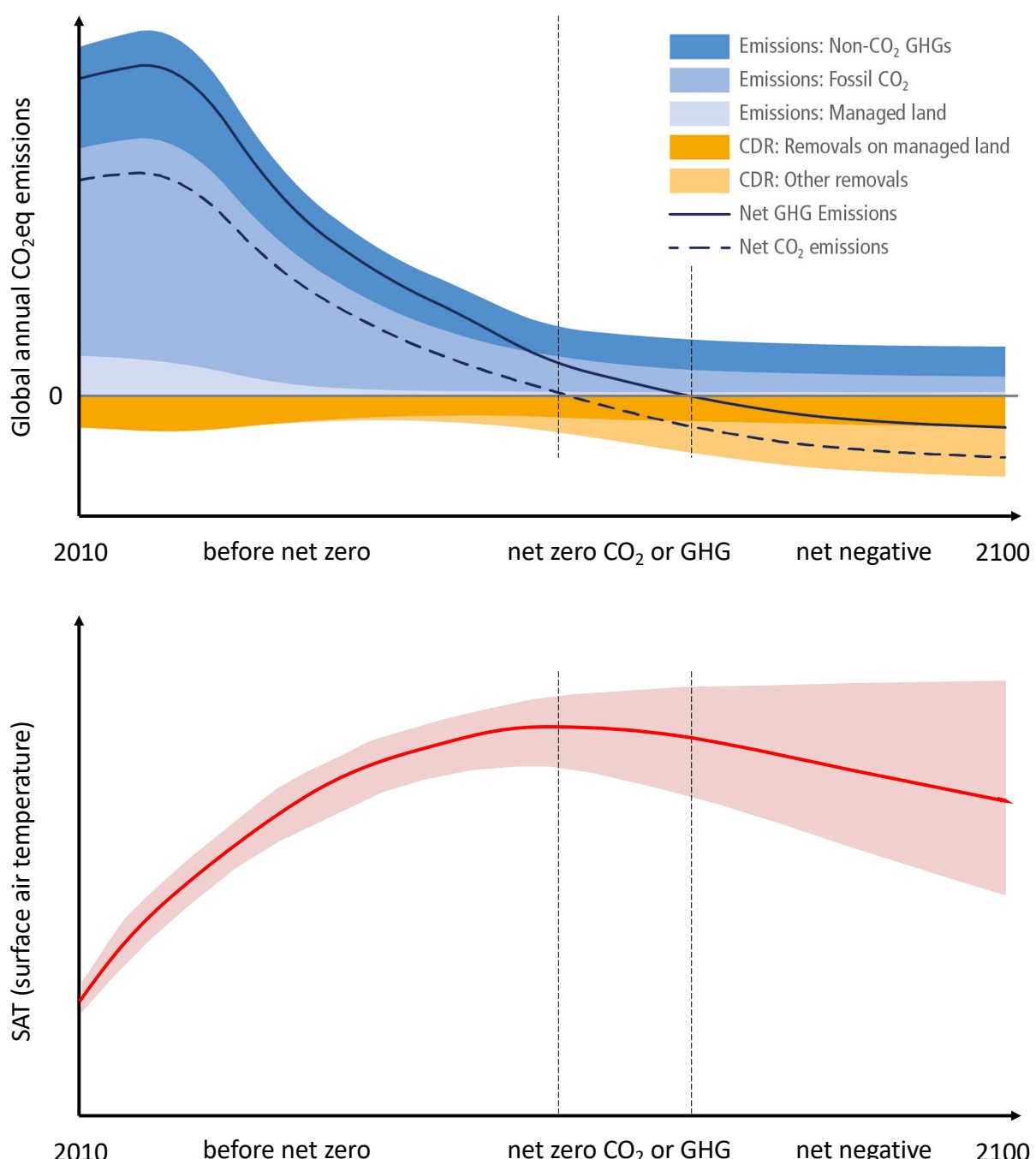

**Figure 1: Top: The role of CO₂ removal ("CDR") in a stylised pathway of ambitious climate. Dark orange illustrates CO₂ removals from land management and light orange illustrates removal from other CDR methods, including ocean-based methods. Note that net-zero CO₂ is reached well before net-zero greenhouse gas (GHG), and the the amount of CDR required for net-zero CO₂ can be substantially smaller than the amount of CDR required for non-zero GHG. Any contribution of Ocean Alkalinity Enhancement**

490

would be included in 'CDR: Other removals'. Modified from IPCC (2022, Cross-Chapter Box 8, Figure 2). Bottom: The corresponding global surface air temperature, with shading indicating a typical uncertainty range.

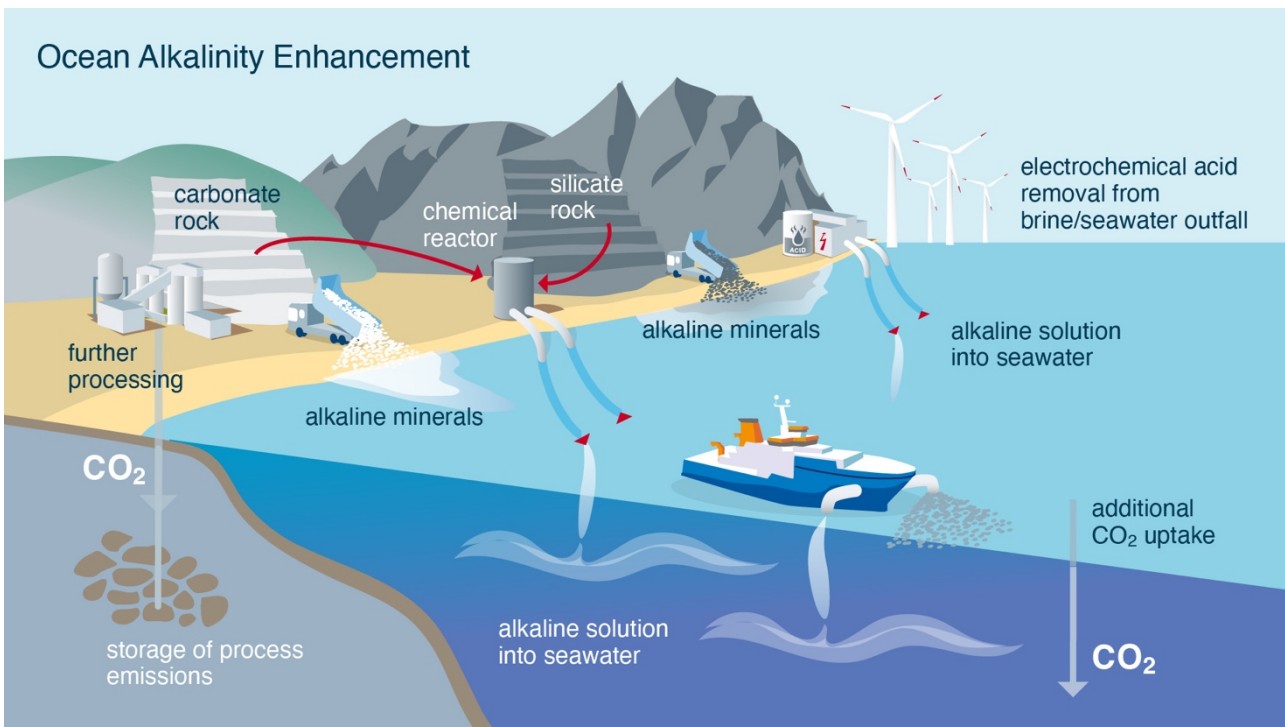

495

**Figure 2: Illustration of various methods that have been proposed as Ocean Alkalinity Enhancement measure to achieve Carbon Dioxide Removal.**