# Peer review of "Climate targets, Carbon Dioxide Removal and the potential role of Ocean Alkalinity Enhancement"

_State of the Planet, 2023_

## Referee Comment (RC2)

**Generally, a referee comment should be structured as follows: an initial paragraph or section evaluating the overall quality of the preprint ("general comments"), followed by a section addressing individual scientific questions/issues ("specific comments"), and by a compact listing of purely technical corrections at the very end ("technical corrections": typing errors, etc.)**

General comments

To introduce the OAE best practices guide, this chapter provides the background, motivation and aims of this document, whilst impressing the relevance and timeliness of advancing CDR science. The chapter is well-written and includes references that appropriately situate this text within the conversation of relevant scientific literature.

Section 1 does a good job of providing the salient information for the reader to understand the importance and need of CDR in climate action portfolios to achieve the Paris Agreement goals. However, at times I felt the flow of the arguments and the structure of the paragraphs could be improved for increased clarity, and have made a few suggestions in the "Technical comments" section of my review.

Section 2 establishes the need for 'novel' approaches outside of land-based CDR (i.e. the role the ocean can play) in order to reach net-zero. Importantly, the authors highlight the pressing need to address knowledge gaps in efficacy, risks and benefits, as well as intersecting societal factors. Crucially, the authors make the case for the urgent need for research now, in order to make well-informed decisions in future. At the end of this section, the concept of MRV is introduced. Since this text forms an introduction, and considering the nature of this guide, a short paragraph explaining MRV, its role and general current challenges could be useful here.

Section 3 provides a clear summary of OAE – how it works, where it can be applied and where research on OAE currently sits. Addressing these knowledge gaps provides the motivation for the guide presented in Section 4, which provides an overview of the contents. Section 5 situates the this guide within the context of similar initiatives as well as outlines the project development, protocol for transparency and stakeholder involvement.

Overall, I congratulate the authors on this chapter. Minor specific and technical comments can be found in the attached file.

Specific comments

- Lines 26-28. Quantifying this statement would strengthen it. Why are ambitious reductions required? How close are we already to reaching these thresholds?
- Lines 29-32. The last sentence of this paragraph feels a bit out of place as it is retrospectively explaining the statement in the first sentence in this paragraph of why GHG reductions are needed. I suggest moving the last sentence to the beginning of the

paragraph. If you end the paragraph on the idea suggested in the previous bullet point, it will still tie nicely into the second paragraph.

- Line 40. Define "balance" – do you mean net zero? I think net zero is a more precise term so I would recommend using that instead of balance when possible

- Lines 139-140. "Attractive aspects of OAE compared to many other methods, in particular those that 140 store carbon in biomass, are its potential to reduce ocean acidification at least locally" – can the citation Albright et al 2016 (10.1038/nature17155) be used here? I see it is mentioned later in line 162 but think it could be useful here as well.

- Lines 153-167. This is a minor suggestion. In this paragraph, two types of studies are presented as providing evidence on the effectiveness and impacts of OAE: modelling and experimental studies. While the benefits and limitations of each approach, and how they complement each other, is alluded to in the text, I think these key points could be clarified – i.e. that modelling studies while simplifications of reality can provide large-scale estimates of CDR potential, while small-scale experimental studies give insight into realised effectiveness of alkalinity additions and measuring impacts that cannot be predicted from simplified modelled systems. However crucial knowledge gaps in determining the best method for alkalinity deployment, the optimal alkaline materials to use, etc. limit our ability to accurately predict the carbon storage potential and co-benefits/risks of OAE. This suggested re-organization might transition better into section 4.

Technical corrections

- Line 19. … and marine CDR options are receiving more and more interest
- Line 26. Achieving the Paris Agreement's goal of limiting global warming to well below 2°C above pre-industrial levels
- Line 43. I think this sentence can be interpreted in a way that contradicts a bit with the sentence previous, because not "all" greenhouse gas emissions need to be avoided (i.e. complete stop of any emissions whatsoever). Recommend phrasing such as: "Since it is not foreseeable that this can be achieved through reducing GHG emissions alone, …"
- Line 46. I was a bit confused as to why the narrative transitioned from talking about CO2 removal to non-CO2 greenhouse gas removal. I had to read this sentence several times to understand it. For more easy reading, I suggest starting with something like: "Even in scenarios with very aggressive CO2 emissions reduction, it is likely CDR will still be necessary to compensate for the emissions of industries that are difficult to de-carbonize (e.g. cement production, etc), or for non-CO2 greenhouse gas for which no viable large-scale removal technologies presently exist." I was also going to recommend adding a citation here, but then all the support from literature appears in the next paragraph. Consider even cutting this paragraph at "Since it is not foreseeable …" and then joining this with the following paragraph.
- Line 61. First 2 sentences here read almost as a concluding/summary sentence for previous paragraph

- Line 87. The way this is worded makes it seem like all EBS are terrestrial. Perhaps clarify as "land-based EBS"? Because I do not think it is intended to include coastal blue carbon EBS also in this category
- Line 112. To transition to the next section, and declare the focus of the document, perhaps provide a concluding sentence about how here you choose to focus on a particular abiotic method, OAE, because [...]
- Line 207. Since the amount is reported in US dollars, I recommend using the US convention of 170,000 rather than 170.000
- Lines 339-343 (Figure 1 caption). I think the text in the figure caption does not appear to accurately match the figure. For example "dark green" is referenced but I do not see dark green anywhere on the figure. Please check.

---

## Author Comment (AC1)

Reply to comment of Anonymous Review #1

*Reviewer comment:*

*The manuscript by Oschlies et al. is the introductory chapter in a Best Practices Guide to OAE Research. The full guide will contain seven chapters, which compare and synthesise previously published methods, and offer guidance for future research. Given that Oschlies et al. have only written the introduction chapter it does not present new results or new research. The manuscript is very well written and presents a comprehensive overview of the need for climate dioxide removal (CDR), and the role the ocean can, or should, play. The description of ocean alkalinity enhancement (OAE) is detailed and comprehensive, yet the nuances are easy to follow and understand. The referencing is comprehensive, and I have not identified key publications that have not been, but should be, referenced here. I commend the authors on their overview.*
*The only issue, and the reason I suggest minor revisions, is about the timeline of our climate goals and how this affects the realism of the suggested CDR methods. The authors correctly state that the goals are to reach net-zero emissions by mid-century. We are now in 2023 so mid-century is quite close in time. Yet the technology necessary for CDR, and marine CDR in particular is in its infancy or non-existent (as noted by the authors of this manuscript).*
*I have just read the introductory chapter so this topic may be covered elsewhere in the Best Practices Guide. But a (brief) section should be added to discuss the necessary timeline, and how realistic/unrealistic it is to achieve operational technology and methods to successfully implement marine CDR. Even if covered elsewhere this aspect deserves mention in the introductory chapter.*

Response:

We thank Anonymous Referee #1 for the positive and supportive comment and the suggestion to include a brief section to discuss the necessary timeline for eventual implementation should OAE ever be deployed in a manner helpful for meeting current climate targets to reach net-zero emissions by mid-century.

We propose the following addition:

[revised manuscript text omitted]

---

## Author Comment (AC2)

Reply to comment of Anonymous Review #2.

Reviewer comments are displayed in italics, our responses in roman font.

*Reviewer comment:*

**General comments**
*To introduce the OAE best practices guide, this chapter provides the background, motivation and aims of this document, whilst impressing the relevance and timeliness of advancing CDR science, and the role OAE can play. The chapter is well-written and includes references that appropriately situate this text within the conversation of relevant scientific literature.*
*Section 1 does a good job of providing the salient information for the reader to understand the importance and need of CDR in climate action portfolios to achieve the Paris Agreement goals. However, at times I felt the flow of the arguments and the structure of the paragraphs could be improved for increased clarity, and have made a few suggestions in the "Technical comments" section of my review.*
*Section 2 establishes the need for 'novel' approaches outside of land-based CDR (i.e. the role the ocean can play) in order to reach net-zero. Importantly, the authors highlight the pressing need to address knowledge gaps in efficacy, risks and benefits, as well as intersecting societal factors. Crucially, the authors make the case for the urgent need for research now, in order to make well-informed decisions in future. At the end of this section, the concept of MRV is introduced. Since this text forms an introduction, and considering the nature of this guide, a short paragraph explaining MRV, its role and general current challenges could be useful here.*
*Section 3 provides a clear summary of OAE – how it works, where it can be applied and where research on OAE currently sits. Addressing these knowledge gaps provides the motivation for the guide presented in Section 4, which provides an overview of the contents. Section 5 situates the this guide within the context of similar initiatives as well as outlines the project development, protocol for transparency and stakeholder involvement.*
*Overall, I congratulate the authors on this chapter. Minor specific and technical comments can be found in the attached pdf file.*

We thank reviewer #2 for the supportive evaluation and very helpful and constructive comments! All recommendations and suggestions in the 'Specific comments' and 'Technical comments' are much appreciated and addressed individually below. A short paragraph explaining MRV, its role and general current challenges will be included in section 2 as follows:

'A particular challenge for marine CDR concerns monitoring and verification of any CDR-induced carbon fluxes and carbon storage, essential for appropriate carbon crediting. Detection and attribution of OAE signals is particularly challenging due to the large natural marine carbon pool that already contain a considerable anthropogenic perturbation, their high resolution temporal and spatial variability and the spatial and temporal decoupling of air-sea $CO_2$ fluxes and carbon storage in the interior. The determination of a baseline, of the additional carbon sequestered, and of its durability will likely be associated with considerable uncertainties. A key aspect of Monitoring, Reporting and Verification (MRV) is the development of transparent schemes that allow a reliable determination of OAE itself, and of

consequent impacts on the carbon cycle and hence climate, as well as the association of carbon credits with individual OAE activities.'

*Specific comments*
- *Lines 26-28. Quantifying this statement would strengthen it. Why are ambitious reductions required? How close are we already to reaching these thresholds?*

  We will add that 'Achieving the Paris Agreement…can thus be converted to a remaining carbon budget that, for current global emissions, will be used up in a few years for the 1.5°C target and about 2 decades for the 2°C target (United Nations Environmental Programme, 2022).

- *Lines 29-32. The last sentence of this paragraph feels a bit out of place as it is retrospectively explaining the statement in the first sentence in this paragraph of why GHG reductions are needed. I suggest moving the last sentence to the beginning of the paragraph. If you end the paragraph on the idea suggested in the previous bullet point, it will still tie nicely into the second paragraph.*

  We agree and thank the reviewer for this suggestion and will change the sequence of sentences accordingly.

- *Line 40. Define "balance" – do you mean net zero? I think net zero is a more precise term so I would recommend using that instead of balance when possible*

  The term 'balancing' was introduced by the Paris Agreement and we therefore want to keep it. We will add 'i.e. net zero' after 'achieve a balance'.

- *Lines 139-140. "Attractive aspects of OAE compared to many other methods, in particular those that store carbon in biomass, are its potential to reduce ocean acidification at least locally" – can the citation Albright et al 2016 (10.1038/nature17155) be used here? I see it is mentioned later in line 162 but think it could be useful here as well.*

  Thanks, good point. The reference will be added here.

- *Lines 153-167. This is a minor suggestion. In this paragraph, two types of studies are presented as providing evidence on the effectiveness and impacts of OAE: modelling and experimental studies. While the benefits and limitations of each approach, and how they complement each other, is alluded to in the text, I think these key points could be clarified – i.e. that modelling studies while simplifications of reality can provide large-scale estimates of CDR potential, while small-scale experimental studies give insight into realised effectiveness of alkalinity additions and measuring impacts that cannot be predicted from simplified modelled systems. However crucial knowledge gaps in determining the best method for alkalinity deployment, the optimal alkaline materials to use, etc. limit our ability to accurately predict the carbon storage potential and co-benefits/risks of OAE. This suggested re-organization might*

*transition better into section 4.*

We follow this helpful suggestion to improve the clarity and structure of the manuscript and will amend the section accordingly.

**Technical corrections**
We thank the reviewer for these detailed and helpful corrections!
- *Line 19. ... and marine CDR options are receiving more and more interest*

  done

- *Line 26. Achieving the Paris Agreement's goal of limiting global warming to well below 2°C above pre-industrial levels*

  done

- *Line 43. I think this sentence can be interpreted in a way that contradicts a bit with the sentence previous, because not "all" greenhouse gas emissions need to be avoided (i.e. complete stop of any emissions whatsoever). Recommend phrasing such as: "Since it is not foreseeable that this can be achieved through reducing GHG emissions alone, ..."*

  done

- *Line 46. I was a bit confused as to why the narrative transitioned from talking about CO2 removal to non-CO2 greenhouse gas removal. I had to read this sentence several times to understand it. For more easy reading, I suggest starting with something like: "Even in scenarios with very aggressive CO2 emissions reduction, it is likely CDR will still be necessary to compensate for the emissions of industries that are difficult to de-carbonize (e.g. cement production, etc), or for non-CO2 greenhouse gas for which no viable large-scale removal technologies presently exist." I was also going to recommend adding a citation here, but then all the support from literature appears in the next paragraph. Consider even cutting this paragraph at "Since it is not foreseeable ..." and then joining this with the following paragraph.*

  We have re-arranged this sentence as follows:
  'Therefore, carbon dioxide removal (CDR) will likely have to balance not only hard-to-abate residual emissions of CO2, e.g. from cement production, waste incineration, aviation and maritime transport, but also those of non-$CO_2$ GHGs, in particular from agriculture.'

- *Line 61. First 2 sentences here read almost as a concluding/summary sentence for previous paragraph*

  Good point, we moved these 2 sentences to the previous paragraph.

- *Line 87. The way this is worded makes it seem like all EBS are terrestrial. Perhaps clarify as "land-based EBS"? Because I do not think it is intended to include coastal blue carbon EBS also in this category*

  Yes, thanks, good point. Added 'terrestrial ecosystem-based solutions'.

- *Line 112. To transition to the next section, and declare the focus of the document, perhaps provide a concluding sentence about how here you choose to focus on a particular abiotic method, OAE, because [...]*

  Thanks for this suggestion to improve the text. We now refer to the assessment provided by the report of the National Academies (NASEM, 2021) and add: 'Among marine CDR methods investigated, abiotic approaches have been assessed as those with the lowest knowledge base and highest efficacy (NASEM, 2021). Improving their knowledge base therefore appears prudent, and we here concentrate on ocean alkalinity enhancement.'

- *Line 207. Since the amount is reported in US dollars, I recommend using the US convention of 170,000 rather than 170.000*

  done

- *Lines 339-343 (Figure 1 caption). I think the text in the figure caption does not appear to accurately match the figure. For example "dark green" is referenced but I do not see dark green anywhere on the figure. Please check.*

  Uups, sorry and thanks for pointing this out. We had switched figure formats from an earlier version back to the original IPCC color scale without adapting the figure caption. Now corrected.

---

## Author Comment (AC3)

Reviewer comments are displayed in italics, our responses in roman font.

**Reviewer comment:**

*This MS provides the introduction to a new Best Practices Guide for research on ocean alkalinity enhancement (OAE), giving the background to the development of that document. It gives the climate policy context for ocean-based methods for carbon dioxide removal, with specific focus on OAE.  It is well-structured and clearly written, covering the main science issues and knowledge gaps that are to be later discussed in greater detail.*
*The only significant gap would seem to be (brief) consideration of the current status of OAE governance, primarily from a regulatory perspective at international level – since this was considered to be a very high constraint on the OAE feasibility by IPCC (Bindoff et al., 2019; Fig 5.23).  The authors may consider such issues to be out of scope for the guide; however, that would not seem to be the case, since substantive subsequent content is indicated ("The guide also discusses the legal context in which research occurs"; line 179).*
*The nature of decision-making at UN bodies is relevant here. Since formal decisions are based on consensus and agreement, a cautionary approach is the most usual outcome – as is evident by decisions to date by the Convention on Biological Diversity (CBD) and the London Convention/London Protocol (LC/LP) on 'marine geoengineering' in general and ocean fertilization in particular.  A major concern by CBD and LC/LP parties is the risk of adverse transboundary effects, with the actions of one nation state negatively affecting another; such effects may be unlikely for OAE, nevertheless, they could occur even if OAE deployments are limited to territorial waters.  For climate-scale OAE, there would also need to be international agreement on carbon accounting within the UNFCCC framework, a topic that is likely to be highly contentious (and therefore taking a very long time to resolve). .*
*It is relevant that the LC/LP has recently identified OAE as an approach requiring further attention: https://www.imo.org/en/MediaCentre/PressBriefings/pages/Marine-geoengineering.aspx.  GESAMP (2019) could also usefully be cited in the context of governance issues.*

*RESPONSE: We thank the reviewer for flagging these issues. We agree that they require consideration and have addressed them extensively later in the guide (in the chapter on "Legal Considerations" of the OAE guide 23). We proposed to add further discussion of the governance issues in the introduction (see our response to reviewer 1) and then refer readers to the chapter on Legal Considerations. We now also refer to the GESAMP 2019 report.*

**A few minor comments:**
- *Line 59: "several Gt CO2 per year globally".  It would be helpful to be more specific regarding the amount of residual emissions that are included in IPCC scenarios (presumably 2-3 Gt pa, on the basis of "close to 20%").*

  done: We have modified the sentence to read:
  'between 10% and  20% of today's emissions, i.e. about 6 to 12 Gt $CO_2$e per year globally), where $CO_2$e includes the $CO_2$ equivalents of non-$CO_2$ GHGs that are estimated to contribute half to two thirds of the residual emissions (Buck et al.,

2023).'

- *Line 65:  Change "current global CDR deployment" to "current CO2 removal" (since this is 'unintentional' rather than purposeful CDR).*

  done

- *Line 108: After "macrophytes", insert "(e.g. seaweed)"; that is more understandable.*

  done

- *Line 114: "with high (> Gt CO2 yr-1 scale) theoretical sequestration".  It would be more informative if actual estimates of maximum CO2 removal can be given here, by several authors; e.g.: "between 3 -30 Gt CO2 yr-1 theoretical sequestration (Kohler et al. 2013; Renforth & Henderson, 2017; Feng et al 2017").*

  We thank the reviewer for pointing this out! We have accordingly changed the text to: 'with high theoretical sequestration potential in the range of 3 to 30 Gt $CO_2$ $yr^{-1}$ (Köhler et al., 2013; Renforth and Henderson, 2017; Feng et al., 2017)'

- *Line 140:  What is meant by "at the expense of imperfect CDR"?  Explain – or delete.*

  We wanted to say that addition of alkalinity without equilibration with atmospheric CO2, i.e. imperfect CDR, reduces acidification more than alkalinity addition with CO2 equilibration. However, as even complete equilibration of added alkalinity with atmospheric CO2 will lead to a small increase in pH and thus a small reduction of acidification, we decided to delete the phrase.

- *Line 185-6:  "We have to widen the space of options… climate targets" seems rather wordy - and too prescriptive.  Simplification and linkage with previous sentence is suggested: "… is urgently needed, to enable society to define and design appropriate actions to reach agreed climate goals".*

  done, much appreciated.

*EXTRA REFERENCES CITED ABOVE*
*Bindoff, N.L. et al (2019) Changing Ocean, Marine Ecosystems, and Dependent Communities. Chapter 5 in: IPCC Special Report on the Ocean and Cryosphere in a Changing Climate [eds H.-O. Pörtner et al]. Cambridge University Press.*

*Feng EY et al (2017). Model-based assessment of the CO2 sequestration potential of coastal ocean alkalinization. Earth's Future, 5(12): 1252-1266.*

*Köhler P et al (2013) impact of open ocean dissolution of olivine on atmospheric CO2, surface ocean pH and marine biology. Environmental Research Letters, 8(1): 014009.*

GESAMP (2019) High level review of a wide range of proposed marine geoengineering techniques". (Boyd, P.W. and Vivian, C.M.G., eds.). (IMO/FAO/UNESCO-IOC/UNIDO/WMO/IAEA/UN/UN Environment/ UNDP/ISA Joint Group of Experts on the Scientific Aspects of Marine Environmental Protection). Rep. Stud. GESAMP WG 41, Reports & Studies Series.

---

## Author Comment (AC4)

Reviewer comments are displayed in italics, our responses in roman font.

**Reviewer comment:**

*I commend the authors for writing a concise and informative introduction to their Best Practices Guide For Ocean Alkalinity Enhancement Research, and for organizing the production of a guide that will undoubtedly prove useful for researchers, as well as practitioners, in this nascent field. My comments and suggestions on this introductory chapter are below.*

We thank the reviewer, Justin Ries, for his helpful and constructive review and its positive assessment.

*Line 29: 'net-zero requirement for avoiding further temperature rise' – should clarify that this is 'further temperature rise beyond the 1.5 – 2 deg IPCC target' (not beyond the present-day mean global temperature)*

This sentence has been rewritten.

*Line 34: The discussion and corresponding figure showing how to achieve IPCC's target of net zero CO2 emissions by 2050 is useful for illustrating the need for CDR in addition to emissions reductions, but it would be helpful (and potentially compelling for skeptics) to see the corresponding mean atmospheric pCO2 and mean global temperatures that correspond to that target emissions trajectory, since those relationships are the basis of the authors' rationale for pursuing CDR in the first place. Otherwise, the non-specialist may not grasp what achieving 'net zero by 2050' means in terms of global climate change. Perhaps include another panel above the net zero figure showing the corresponding changes in mean global pCO2 and temperature over the same interval.*

Good point. We will modify the figure and consider adding a panel with the corresponding temperature changes.

*Line 77: should clarify here and elsewhere whether the 2/1.5 deg C warming target is relative to pre-Industrial mean global temp or present-day mean global temp.*

Thanks, this clarification is now included.

*Line 100: The authors should differentiate between particulate inorganic carbon and dissolved inorganic carbon when referencing 'inorganic carbon' in this sentence (presumably they are referring only to DIC): 'The ocean holds more than 50 times as much inorganic carbon (in the form of dissolved inorganic carbon) as the pre-industrial atmosphere'*

Thanks, this is now clarified.

*Line 104: should 'marine CDR' instead be 'marine CDR by OAE' here?*

In this section 2, we still refer to all marine CDR and only focus on OAE in section 3. Therefore we decided to leave the statement as is.

*Line 125: should differentiate between the time needed for pCO2 of air and seawater to fully vs. partially equilibrate. It is true that air/sea can take years to fully equilibrate, but the equilibration will be an inverse exponential function of time, meaning a disproportionate share of the equilibration will occur in the beginning of the equilibration interval. This is an important but often overlooked distinction that has important implications regarding the perceived challenges of quantifying CDR by OAE.*

Good point, thanks! The text has now been reformulated to read 'Air-sea gas equilibration of $CO_2$ can take months to years (Jones et al., 2014) and may pose specific challenges to MRV (He and Tyka, 2023). However, along the path to equilibration, air-sea $CO_2$ fluxes approach zero following, for otherwise constant environmental conditions, an inverse exponential function, and a disproportionate share of the total $CO_2$ flux typically occurs at the beginning of the equilibration period.'

*Line 131: It seems that the definition of marine CDR should be expanded from 'OAE qualifies as marine CDR if CO2 is transferred directly from the atmosphere into seawater' to 'OAE qualifies as marine CDR if CO2 is transferred from the atmosphere or seawater into stable carbonate or bicarbonate ions in seawater', as both processes will result in the eventual drawdown of CO2 from the atmosphere. Otherwise, we will miss an important and efficient pathway in CO2 removal. Likewise, CO2 removal from the atmosphere alone is not sufficient for CDR, as increasing the pCO2 of the atmosphere through increased CO2 emissions would increase the flux of CO2 from the atmosphere to the ocean, but surely this should not constitute marine CDR (as the alleviation of the atmospheric CO2 pressure would cause off-gassing of the dissolved CO2 back to the atmosphere unless balanced by alkalinity addition). This point is also illustrated by the author in an earlier paragraph, where they state: 'Alkalinity enhancement results in the consumption of protons, a corresponding increase in the pH, which results in a decrease of the partial pressure of CO2 in seawater. If applied to the surface ocean, and depending on the initial air-sea CO2 gradient, it would promote CO2 uptake from - or lessen CO2 release to - the atmosphere, in both cases leading to a net reduction in atmospheric CO2 at the expense of an increase in the oceanic carbon pool.' In the case that the flux of CO2 from the ocean to the atmosphere is lessened by OAE, this would not satisfy the authors' current requirement that 'CO2 is transferred directly from the atmosphere into seawater', but would instead reduce the rate that seawater CO2 is released to the atmosphere, thereby resulting in a theoretical 'net reduction in atmospheric CO2' – which should qualify the activity as successful marine CDR. Perhaps a more useful framing for CDR is the transfer of C from shorter residence time reservoirs (atmospheric CO2, seawater CO2, terrestrial biomass, marine biomass in mixed layer etc.) to longer residence time reservoirs (bicarbonate ion reservoir, carbonate ion reservoir, terrestrial and marine biomass transported to deep ocean below mixed layer, etc.) (c.f., Prentice, I. C., 2001, The carbon cycle and atmospheric carbon dioxide. Climate change 2001: the scientific basis, Intergovernmental panel on climate change. hal-03333974) or, more colloquially, transferring C from the 'fast C cycle' to the 'slow C cycle', as this encompasses the ultimate goal of marine CDR – i.e., net reduction of atmospheric CO2, regardless of the strict and not necessarily relevant 'transfer of CO2 between ocean and atmosphere'.*

Many thanks for this comment! We agree that the wording was sloppy and that CDR has to be defined via 'additional' CO2 that is removed from the atmosphere, and into the ocean in the case of marine CDR. We do not agree that CO2 removal from seawater automatically qualifies as CDR, which is only the case when a CO2 flux from the atmosphere to the ocean is induced. We have added the word 'additional' to exclude the CO2 that transfers naturally from the atmosphere to the ocean from counting as CDR.

We decided to separate the discussion of residence times from the definition of CDR, as there is a continuum of residence times (e.g. Siegel et al., ERL, 2021) and with a risk of introducing ambiguities when partitioning into 'slow' and 'fast'.

*Line 143: 'leakage' by OAE-induced precipitation (or reduced dissolution) of CaCO3 at the seafloor (or really anywhere in the water column below the mixed layer) is probably not relevant over climate-relevant timescales because it will take 100-1000s of years for those waters to return to the surface and re-equilibrate with (i.e., offgas CO2 to) the atmosphere, just as shoaling of the carbonate compensation depth in response to CO2-induced OA will not sequester anthropogenic CO2 over fast enough timescales to prevent warming (hence, the bind we are in).*

We agree that the leakage referred to here is a slow process, but time scales of 100s or 1000s of years are still climatically, and hopefully societally, relevant and need to be considered in decisions made today. We have added the following text to describe the situation in a more comprehensive way:

'Possible leakage effects via impacts of OAE on pelagic calcifiers are uncertain (Bach et al., 2019), and feedbacks via changes in dissolution and preservation of carbonates on the sea floor operate on timescales of hundreds to thousands of years (e.g. Gehlen et al., 2008). While there is little indication that leakage is a major concern for OAE on shorter than centennial timescales, a quantitative assessment of leakage across the spectrum of timescales is lacking. '

with references to

- L. T. Bach, S. J. Gill, R. E. M. Rickaby, S. Gore, and P. Renforth. CO2 removal with enhanced weathering and ocean alkalinity enhancement: Potential risks and co-benefits for marine pelagic ecosystems. Frontiers in Climate, 1:7, 2019.
- M. Gehlen, L. Bopp, and O. Aumont. Short-term dissolution response of pelagic carbonate sediments to the invasion of anthropogenic CO2: A model study. Geochemistry Geophysics Geosystems, 9, 2008.

*Line 150: should probably add 'so long as any CO2 emitted in their production (e.g., Ca(OH)2 or Mg(OH)2 produced through calcination of CaCO3 or MgCO3, respectively) is accounted for'*

We added the sentence 'Employing these for OAE would require proper accounting of any $CO_2$ emitted in their production (e.g., $Ca(OH)_2$ or $Mg(OH)_2$ produced through calcination of $CaCO_3$ or $MgCO_3$, respectively).'

*Line 161: Was Albright et al (2016) the first ocean acidification field experiment (see Hall-Spencer et al. 2008 field experiments using volcanic vents, etc.)? Or just the first field experiment to modify seawater pH through alkalinity addition rather than direct pCO2 manipulation? May also be worth mentioning that insight into impact of OAE on marine organisms can be gained from past research by the shellfish industry investigating the utility of so-called 'sweetening' the water through addition of mainly soda ash (Na2CO3), a practice utilized in shellfish hatcheries for decades, and also in the academic and industrial fields of 'river liming', which dissolved primarily CaCO3 and dolomite in higher latitude watersheds to offset the effects of acid rain (due to NOx and SOx emissions) in the 1960s and 1970, but is still practiced today in Canada and some Scandinavian countries, among other places.*

Thanks, very good points! We rephrased the Albright et al. (2016) experiment as 'first OAE field experiment carried out in the context of ocean acidification research ' as this was, to our knowledge, the first experiment where alkalinity was added for a scientific experiment in the field.
We also took up the reviewer's suggestion and added the following text: 'Insight into possible impacts of OAE on marine organisms can be gained from past research by the shellfish industry investigating the utility of so-called 'sweetening' the water through addition of mainly soda ash ($Na_2CO_3$), a practice utilized in shellfish hatcheries for decades, and also in the academic and industrial fields of 'river liming', which dissolved primarily $CaCO_3$ and dolomite in higher latitude watersheds to offset the effects of acid rain in the 1960s and 1970, but is still practiced today in Canada and some Scandinavian countries, among other places. '

*Line 173: need more concise phrasing than 'enhancing technological readiness'; perhaps 'developing (or implementing) scalable methodologies'*

Thanks. We added 'start-ups working on enhancing technological readiness and developing scalable methodologies'

*Line 184: perhaps change 'in a situation where' to 'at a time when'*

done

*Misc:*

*Confirm whether 'Ocean Alkalinity Enhancement' and 'Carbon Dioxide Removal' should be capitalized in title and throughout manuscript.*

Agreed and capitalized throughout the text.

*Use of term 'monitoring, reporting and verification' in abstract without defining the term may be confusing to readers, as the phrasing really only has meaning when the three terms are defined and understood in aggregate.*

Agreed that this might be confusing, in our view primarily the 'reporting' part. Monitoring and verification should be self-explanatory, and we have kept these two terms in the abstract. In addition, there is now a new more detailed section defining MRV included in the manuscript.

*Respectfully submitted,*
*J. Ries*